# Chronic Stress-Related Gastroenteric Pathology in Cheetah: Relation between Intrinsic and Extrinsic Factors

**DOI:** 10.3390/biology11040606

**Published:** 2022-04-15

**Authors:** Sara Mangiaterra, Laurie Marker, Matteo Cerquetella, Livio Galosi, Andrea Marchegiani, Alessandra Gavazza, Giacomo Rossi

**Affiliations:** 1School of Biosciences and Veterinary Medicine, University of Camerino, 62024 Matelica, Italy; matteo.cerquetella@unicam.it (M.C.); livio.galosi@unicam.it (L.G.); andrea.marchegiani@unicam.it (A.M.); alessandra.gavazza@unicam.it (A.G.); giacomo.rossi@unicam.it (G.R.); 2Cheetah Conservation Fund, Otjiwarongo 9000, Namibia; director@cheetah.org

**Keywords:** cheetah, gastritis, stress-related disease, immune response

## Abstract

**Simple Summary:**

The cheetah is the fastest land mammal. Habitat destruction, high mortality due to other predators, and illegal wildlife trade has led to a decrease in the wild population. Currently, the global adult population present in their natural habitat is estimated to be 7100 individuals. In captivity, the population suffers from limited reproduction and disease. Both the wild and captive populations have reduced genetic diversity from a historic bottleneck, leading to increased ecological and environmental vulnerability. Over the years, conservation programs have been developed for habitat protection, management of human–animal conflict, and the study of disease and genetics. Among these, the one with the greatest prevalence in captivity is chronic gastritis. This review analyzes the scientific literature on gastric pathology in cheetah, with the potential causes divided into “extrinsic factors”, such as living conditions and diet, and “intrinsic factors”, including the presence of *Helicobacter*-like organisms and the genetic predisposition.

**Abstract:**

The rapid decline of cheetah (*Acinonyx jubatus*) throughout their range and long-term studies of captive breeding has increased conservation action for this species including the study of chronic diseases. Gastritis is one of the captive diseases that leads to high mortality presented with symptoms including vomiting, diarrhea, anorexia, and weight loss. The disease presents different histological lesions in the gastrointestinal tract that are characterized by inconstant and different clinical appearance in captive and free-range cheetahs. The aim of this review is to summarize the causes of chronic gastritis in the cheetah. Factors including diet, living conditions, infections with gastric *Helicobacter*-like organisms (GHLOs), the lack of genetic polymorphism and the cheetah’s specific-immunocompetence are analyzed. All studies on gastroenteric cheetah pathologies, conducted between 1991 (to the best of our knowledge, the first report on online databases) and 2021, are included in this review, highlighting the possible correlation between stress-related captive conditions and chronic gastric pathology.

## 1. Introduction

The cheetah (*Acinonyx jubatus*) is the world’s fastest land mammal, and the only living species of the genus *Acinonyx*. Once widespread in much of sub-Saharan Africa through Asia, today is classified as “vulnerable” by the International Union for Conservation of Nature (IUCN) [1]. Habitat destruction, conflict with other predators, human–wildlife conflict, illegal wildlife trade, and the lack of genetic diversity from a historic bottleneck, have endangered the survival of this species [1]. In captivity, a high incidence of chronic diseases has been identified and studied [2].

Between 1989 and 1992, the first research on diseases affecting cheetahs in captivity was conducted by L. Munson on 31 adults and 9 cubs in 16 United States zoos to document the incidence of different health conditions [2]. Among these conditions, such as renal diseases, feline infectious peritonitis, testicular degeneration, and pneumonia in cubs, chronic gastritis was found in 91% of cheetahs. Of these cheetahs, 95% of the gastritis cases were caused by spiraliform bacteria (*Helicobacter* spp.) [2,3]. In 1993, an epizootic gastritis, characterized by lymphoplasmacytic infiltrate, was described in a group of cheetahs at the Columbus Zoo (Powell, OH, USA), and was associated with *Helicobacter* species [4]. To date, gastrointestinal diseases represent a high prevalence of recorded cases associated with significant mortality in captive cheetahs [5,6]. Comorbidity of diseases associated with chronic gastritis include Barrett’s esophagus [7], gastro-esophageal reflux disease (GERD), acquired hiatal hernia, and related secondary conditions such as ab ingestis pneumonia [2,8,9], systemic amyloidosis [10], and β-amyloid (Aβ) deposits and neurofibrillary tangles similar to the one observed in Alzheimer’s disease [11]. Clinical signs described in cheetahs with chronic gastro enteric disorders include regurgitation, vomiting, diarrhea with undigested material, and weight loss. Research has shown that the gastrointestinal tract is particularly responsive to different stressors [12]. The relationship between the incidence of gastrointestinal disease in cheetahs and the prevalence of clinical symptoms have been linked to stress conditions of captive animals, possibly due to the inability to perform natural physiological behaviors of the species or to the personality of the individual cheetahs [13]. This article reviews the studies on gastro-enteric disorders in cheetah in order to discuss all factors implicated in the pathogenesis.

## 2. Materials and Methods

### Literature Search Strategy and Study Selection

This retrospective study was conducted with a comprehensive literature review. The online PubMed and ScienceDirect databases were used to conduct the literature search. To perform the studies about gastroenteric disorders in cheetah, a combination of descriptors was used: (“gastritis” OR “gastroenteric” OR “gastroenteric disorder” or “*Helicobacter*”) AND (“cheetah” OR “cheetahs” OR “*Acinonyx jubatus*”). The inclusion criteria included English-language articles that provided descriptions of gastroenteric pathology in cheetahs. Studies concerning only therapy for the treatment of gastrointestinal disease in cheetah or studies involving other animal species or humans were excluded.

## 3. Results

After the application of the criteria and the removal of duplicates, 10 eligible studies were identified (Table 1). According with this literature research, the studies about gastrointestinal disease of captive and wild cheetahs were conducted, between 1991 and 2015, without distinction of subspecies.

## 4. Discussion

The cheetah is the only living species of the genus *Acinonyx*, characterized by peculiar morphological and physiological characteristics [17,18], with the current population estimated at 7100 adult and adolescents in the wild [1] and approximately 1800 animals in captivity [19]. The survival of this species is linked to many aspects including the high incidence of chronic diseases. Among chronic disorders, chronic gastrointestinal diseases, especially gastritis, have been shown to have a high incidence of disease especially in captivity [2].

Captive cheetahs have been shown to have a higher severity of gastritis than free-ranging animals [20]; however, due to the difficulty of monitoring the free-ranging population it is not known to what extent the prevalence to chronic diseases affects the lifespan of the free-ranging population. Species survival has been linked to the immunocompetence that is influenced by genetic factors such as the major histocompatibility complex (MHC) [21]. Studies showed that cheetahs have low genetic variability in the loci of the MHC, although recent studies have shown that this is more evident for MHC-II than MHC-I [22,23], thus predisposing to a decreased function of the phagocytic system.

In captivity, the composition of social groups, size of the enclosure, the number of visitors, and the lack of predatory activity or exercise are associated with stress conditions that predisposes animals to gastritis. In addition, the differences in the diet between captive and free-range cheetahs may lead to nutritional deficiencies and alteration of the cheetah’s microbiome [9]. Extrinsic and intrinsic factors can be identified in this pathological process (Figure 1). Intrinsic factors identified are microbiome and gastric *Helicobacter*-like organisms, immunocompetence and lack of genetic variation. Among the extrinsic factors, captive living conditions and diet are identified.

These factors may cause more evident clinical manifestations and aggravate the disease.

### 4.1. Extrinsic Factors

#### 4.1.1. Captive Living Conditions

Stressors cause an adaptive response of the organism through the interaction between the neuroendocrine and the immune systems [12]. In the course of evolution, vertebrates developed psychophysical and behavioral adaptations in response to harmful or dangerous stimuli [24]. This primary biological response, aimed at survival, is characterized by activation of the hypothalamus–pituitary–adrenal axis, followed by the release of catecholamine from the medullary of the adrenal gland and cortisol from the cortical area [24]. However, the variability in the physiological response to stress in different animal species is not yet fully understood [25]. Early studies have shown that the cause of gastritis in captive cheetahs were linked to *Helicobacter*-like organisms which are found in the wild population, with no signs of disease in the latter [6]. It has been hypothesized that captivity can aggravate stress conditions in cheetah including enclosure size, the presence of the people, diet, the lack of exercise and inability to perform other natural behaviors [13].

The most prevalent disease reported in the cheetah is chronic gastritis associated with increased stress levels [26]. To assess stress in captive and free-ranging cheetahs, cortisol levels were measured using different methods [13,27]. The cortisol concentrations were found to be higher in captive than free-range cheetahs and was associated with an increase in the adrenal gland cortico-medullary ratio [28]. Chronic stress has been linked to a greater release of adreno corticotropic hormone (ACTH), stimulating the cells of the fasciculate area, increasing synthesis and secretion of cortisone, and associating with hypertrophy and hyperplasia of the glandular area [26,27,28]. Age has been linked to potential increases in morphological variation of the adrenal glands and increases the prevalence of chronic diseases [29]. The difference in average life expectancy between free-ranging and captive cheetahs needs to be considered.

In Japan, fecal corticosterone was monitored on seven cheetahs between two seasons (spring to winter) to identify if there was a relationship between the climatic conditions and corticosterone levels in cheetahs. The study showed that cheetahs were susceptible to climatic variations as suggested by higher fecal corticosterone concentrations, caused by the decrease in temperature [30]. This temperature change could be a possible stressor for the cheetahs. 

Stress induces a catecholamine mediated ischemia of the gastric mucosa, with a strong decrease in mucus secretion, also linked to the negative effect of corticosteroids on prostaglandin production [31]. In these conditions there is a destruction of the gastric mucous lining with a decrease in the concentration of bicarbonate and, therefore, an inability to buffer the protons in the stomach [32]. *Helicobacter* infection interfers with the processes of physiological control of gastric acidity. Several clinical studies on humans and animals [33,34,35,36,37] have demonstrated the negative effect that *Helicobacter* (spp.) infection exerts on gastric acid secretion. Physiologically, gastrin-producing G cells and somatostatin-producing D cells regulate gastric acid secretion; they are regulated by a paracrine feedback system of mutual regulation, but also by endocrine and nervous mechanisms [38]. Gastrin, released by G cells, responsible for acid secretion, is influenced by histamine, gastrin, and vagal nerve stimulation [39]. Prostaglandins and nitric oxide are responsible, with the vagal stimulation, of the protective secretion of the layer of gel that covers the gastric mucosa [40]. When there are harmful conditions or pathogens, this protective barrier is destroyed, allowing the acid to spread backwards to the epithelium and cause damage to the mucosa [31]. It has been shown that the gastric colonization of some *Helicobacter* species (for example H. pylori and H. canis) is associated with an increase in the concentration of G cells in the antral mucosa and in gastrin values [35,41]. The infection also decreases the expression of mucosal D cell and somatostatin production. Exposure of canine D cells to TNF-α in vitro reproduces this effect [42]. These changes in gastrin and somatostatin increase acid secretion and lead to ulceration. It is therefore possible that in cheetahs, the effect of stress and/or infection can lead to mucous layer alteration, exposing the mucosa to infection-induced hyperacidity. The inflammation products that are generated further altering acid secretion and leading to gastric atrophy with hypochloridria over time. Additionally, chronic stress induces changes in the neuroendocrine and immune systems, predisposing to chronic diseases as *Helicobacter*-associated gastritis [8].

#### 4.1.2. Diet

As strict carnivores, felids have a gastrointestinal tract characterized by a simple and relatively developed stomach and a small intestine that is predominant in comparison with the large intestine [43,44]. This anatomical conformation has been adapted to the diet of the species based on the consumption of animal tissues. Felids need to take proteins with the diet to obtain the essential amino acids and a non-specific nitrogen source that is used to synthesize other nitrogen compounds [45]. Necessary for metabolism are essential fatty acids, especially linoleic acid and arachidonic acid. Felids are unable to synthesize adequate amounts of arachidonic acid (from linoleic acid) and a food shortage of acid linoleic and arachidonic acid can have negative effects in several systems [46]. Cheetahs have limited delta-6 desaturase [47], an enzyme catalyst for the biosynthesis of polyunsaturated fatty acids (PUFA), that converts linoleic acid *to* gamma-linolenic acid and hence *to* arachidonic acid [48]. This deficiency needs to be supplemented in the diet; however, different concentrations of arachidonic acid are present in the prey they consume; for example, rabbit (Oryctolagus cuniculus) meat contains lower concentrations of arachidonic acid than the meat of ungulates [49]. In addition, depending on the meat preservation process, there may be different concentrations of PUFA. PUFAs’ concentration is elevated in hunted prey in the wild but, in the diet in captive conditions, these acids are highly unstable and rapidly deteriorate on low temperature storage of meats or when carcasses without intrinsic organs are used [50]. At the same time, depending on the type of feeding of the chicken, in chicken (*Gallus gallus**) n*-6 *PUFA* amount is very high [51]. Free-ranging cheetahs hunt ungulates such as kudu (*Tragelaphus strepsiceros*), eland (*Taurotragus oryx*), steenbok (*Raphicerus campestris*), springbok (*Antidorcas marsupialis*), gemsbok (*Oryx gazella*), hartebeest (*Alcelaphus buselaphus*), and other small mammals such as scrub hare *(Lepus saxatilis),* while livestock (cattle -*Bos taurus*- or other members of the *Bovidae* family), goats (*Capra* spp.), and sheep (*Ovis* spp.), are comprised only in a small proportion [52]. In captive cheetahs, the diets are different, being fed with meat from rabbit, chicken, beef, turkey (*Meleagris gallopavo f. domestica*), horse, lamb, and goat [9]. Studies suggested that the prevalence of diarrhea associated with dietary factors was significantly higher in cheetahs fed raw meat or mixed diets (made by the alternation of raw meat, commercially prepared and carcasses), while being lower in carcass-fed cheetahs [9]. This difference in diet suggests that captive cheetah may be susceptible to PUFAs deficiency [48]. Moreover, the ratio between n-3/n-6 PUFAs is an important cofactor in the modulation of the immune system and in the anti-inflammatory function [16]. In fact, long-chain n-3 PUFAs give rise to anti-inflammatory mediators, decrease the production of inflammatory mediators such as eicosanoids and cytokines and the expression of adhesion molecules [51]. It is reported that long chain n-3 and n-6 dietary PUFA ratios could have had immune modulatory and anti-inflammatory effects on the development of gastritis (and related secondary renal amyloidosis) in cheetahs [48,52,53,54]. In addition, studies demonstrate that n-3 PUFAs could reduce *Helicobacter* spp. associated gastric diseases by an inhibitory effect on bacterial growth via disruption of cell membrane leading to bacteria lysis [55], reducing iodoacetamide-induced gastritis by decreasing malondialdehyde (MDA), gastrin, and nitric oxide (NO), and normalizing mucosal glutathione levels [56]. It is well established that *Helicobacter* infection can enhance PGE_2_ synthesis and accelerate n-6 PUFAs metabolism in gastric mucosal cells, which can make the gastric mucosal barrier more fragile. Conversely, since essential dietary fatty acids confer protection to the gastroduodenal mucosa, n-3 PUFAs represent an important gastro-protective factor against damage to the gastric mucosa induced by *Helicobacter*, but also by stress and various categories of drugs [57].

### 4.2. Intrinsic Factors

#### 4.2.1. Microbiome and Gastric *Helicobacter*-like Organisms

The components of diet are essential substrates for the proper functioning of the gastrointestinal apparatus and contribute to the composition of cheetah’s microbiome, conditioning a healthy gut and consequently, the health of the host animal [58]. Cheetah’s GI microbiome is mainly composed by Firmicutes, Fusobacteria, Bacteroidetes, Proteobacteria, and Actinobacteria [59]. The difference in bacterial populations found in the GI microbiome of free-range cheetahs, if compared with captive ones, highlights the presence of bacteria belonging to the classes Clostridia and Erysipelotrichi, possibly associated with the intake of perivisceral fat present in hunted prey [59]. Conversely, *Clostridium chauvoei* and *Enterococcus avium*/*E. hirae* have been detected in captive cheetah and are possibly associated with meat handling by the keepers and the presence of domestic animals [50]. High prevalence of potentially pathogenic bacteria belonging predominantly to the genera Clostridium and *Helicobacter* was found in both captive and free-range cheetah populations [59]. G. Bizzozero (1892) first described the presence of spiral-like microorganisms in the gastric mucosa of dogs (*canis* sp.) and cats (*felis* sp.) [60,61]. A few years later, H. Salomon (1896) demonstrated that similar bacteria could be transmitted to the rat (*Rattus norvegicus*), thus defining their relationship with the epithelium and the signs of cellular and histological anomalies induced by the infection [62]. The first to suggest the potential cytopathogenic effect of gastric spiraliform bacteria on the parietal cells of the stomach was Weber et al. (1958) [63]. The spiraliform bacteria, Gastric *Helicobacter*-like organisms (GHLOs), are responsible for clinical or sub-clinical gastrointestinal diseases. The pathogenesis of GHLOs infection appears related to the bacteria interactions with Toll-like receptors (TLRs), interfering in the regulation of the immune system [64]. Different GHLOs were identified in several mammals including ferrets (*Mustela putorius furo*), domesticated and wild cat (*Felis catus*, and *Felis silvestris catus* respectively), cheetah (*Acinonyx jubatus*) and pig (*Sus scrofa domesticus*.) [65]. The first site of bacterial colonization is gastric antrum where the bacterium adapts easier due to the lower acidity present when compared to the stomach body and bottom [66]. The presence and topographic localization of GHLOs and their association with histological modifications in dogs and cats were compared. Biopsy specimens from the fundus, body, and antrum tracts of the stomach were sampled. In cats, lymphocyte aggregates were found only in GHLOs positive subjects, with localization in the fundus and the body, with a statistically significant association between the presence of GHLOs and chronic gastritis in the body and fundus; in the dog this association is not yet clear [67]. In captive carnivores such as tigers (*Panthera tigris*), lions (*Panthera leo*), cougars (*Puma concolor*), wolf (*Canis lupus*), and hyenas (*Hyaena hyaena*), *Helicobacter*-like organisms’ infection was found in different gastric areas and associated to different histopathological patterns. Severe lesions were found in association with gastric bacteria similar with *H. pylori*, characterized by lymphoplasmacytic and neutrophilic infiltrates [59]. In cheetah, the species of *Helicobacter* identified were *H. heilmanni*, *H. acinonyx* [4,14,15,16,17,18,19,20,21,22,23,24,25,26,27,28,29,30,31,32,33,34,35,36,37,38,39,40,41,42,43,44,45,46,47,48,49,50,51,52,53,54,55,56,57,58,59,60,61,62,63,64,65,66,67,68,69], and H. canis [70]. H. acinonyx is phylogenetically the most similar to H. pylori [8] but, to date, the association between the presence of GHLOs and chronic gastritis remains controversial. Unlike H. pylori, in cheetahs, the isolated GHLOs do not seem to fulfill Koch’s postulates in accordance with which: “*the microorganism must be found in the diseased animal, and not found in healthy animals*”; it “*must be extracted and isolated from the diseased animal and subsequently grown in culture*”; it “*must cause disease when introduced to a healthy experimental animal*” and finally it “*must be extracted from the diseased experimental animal and demonstrated to be the same microorganism that was originally isolated from the first diseased animal*” [71]. In cheetahs, the GHLOs are not found in only sick animals, in this point of view other factors contribute to the development of the gastroenteric pathology having an important role in the pathogenesis of chronic gastritis, related to stress, diet, and individual immune response [8]. The real correlation among the presence of *Helicobacter* spp. and gastrointestinal disease in cheetah has not yet been clearly understood unlike what has been demonstrated in other animals [65].

#### 4.2.2. Immunocompetence and Lack of Genetic Variation

*Helicobacter* spp. infection and the immunocompetence may be associated to chronic-subclinical or clinically evident gastritis in cheetah. Research by O’Brien et al. (1982) reveled that cheetahs went through a historic population bottleneck leaving them genetically monomorphic at the major histocompatibility complex (MHC) [72]. It has been shown that bottlenecked populations have a highly reduced MHC variation associated with low immune adaptability and a prevalence for disease and extinction [73]. The sharp reduction of the population impairs species immune response [74] and the low polymorphisms in MHC I and II genes have been associated with high susceptibility to infectious diseases [72,75]. In MHC, two subgroups are identified: MHC-I expressed on the surface of cells and which are able to present peptides to cytotoxic T-cells; and MHC-II which are present on antigen-presenting cells such as macrophages and lymphocytes, and are able to present processed antigens to T-helper cells [76]. This family of genes encodes receptor molecules to recognize and bind foreign proteins driving the immune cells and the immune response [77]. Pathogens or foreign organisms can enter cells by infection or phagocytosis into specific cells such as macrophages. Through the interaction between protein fragments of the pathogen and the MHC, the activation of the immune response starts [77].

A difference in immuno-competence and seropositivity between captive and free-range cheetahs has been demonstrated [77,78,79]. Chronic diseases were shown to have the highest mortality in cheetahs, with *Helicobacter* spp. gastritis infections causing 95% of cases [79].

Gastritis in cheetah was classified in three degrees, depending on histological severity: gastritis *grade 1*, characterized by moderate inflammation and few necrotic areas interspersed in the epithelial layer; gastritis *grade 2*, characterized by inflammatory infiltrate invading the lamina propria with some glandular dilatations and necrotic areas; gastritis *grade 3*, in which the lamina propria is invaded by inflammatory infiltrate, and the glands appear dilated, with necrotic areas associated to areas of erosion and ulceration [6,7,8,9,10,11,12,13,14,15,16,17,18,19,20,21,22,23,24,25,26,27,28,29,30,31,32,33,34,35,36,37,38,39,40,41,42,43,44,45,46,47,48,49,50,51,52,53,54,55,56,57,58,59,60,61,62,63,64,65,66,67,68,69,70,71,72,73,74,75,76,77,78,79]. The persistence of infection by *Helicobacter* species (such as *H. pylori* and *H. acinonyx*) can be explained by the ability of these bacteria to interact with the host’s immune system. These pathogens are therefore able to manipulate the inflammatory response by inducing a T-type Th1/Th17 helper response. Genetics is a key factor in the control of the immune response and, consequently, in the evolution of the disease. In medium gastritis (or *grade 2*) associated with GHLOs, the immune response is characterized by lymphoid cells similar to that found in *H. pylori* infection in humans [6]. In contrast, in severe gastritis (or *grade 3*) the immuno-pattern is characterized by large numbers of activated B cells and plasma cells [6]. The chronic gastritis to which the cheetah is predisposed brings with it other related diseases suggesting the genetic variation predisposing to the development of the disease. In cheetahs the predominant B cells inflammatory infiltrate characterizes the severe gastritis and indicates a strong imbalance towards a Th2-type response to *Helicobacter* infection.

Two factors could contribute to the imbalance towards the Th2 versus Th1 response: host genetics and poor nitric oxide (NO)/IFN-ɣ production by activated macrophages. The low NO production by macrophages can be related to the inability of the cheetah to synthesize de novo arginine, which is the main NO precursor [80]. This amino acid is also essential for the proliferation of T cells, which is therefore compromised by its depletion [81]. Therefore, the absence in endogenous synthesis of arginine, as in case of felids that showed very low levels of pyrroline-5-carboxylate (P-5-C) synthase and ornithine aminotransferase in the intestinal mucosa, can affect the polarization of Th1-Th2 [82]. As in T cells, arginine influences macrophage’s polarization and metabolic phenotype [83,84]. In the condition of high L-arginine concentration, *naive* CD4 + T cells differentiate into Th1 effector cells secreting higher levels of IFN-γ [85] and inducing a strong macrophages response. Reduced arginine metabolism also negatively influences the nitric oxide synthase inducible isoform (iNOS) expression, reducing the production of NO from L-arginine upon stimulation by proinflammatory cytokines (e.g., interleukin-1, tumor necrosis factor-α, and interferon-γ) [86]. Reduced iNOS activity modifies the host’s immune response, reducing its participation in antimicrobial and antitumor activities as part of the oxidative burst of macrophages [87]. The excessive Th2 response also influence the production of large amounts of immunoglobulin from infiltrating B cells of gastric mucosa, favoring the synthesis of amyloid precursors.

## 5. Conclusions

The cheetah population is currently listed as vulnerable, and the study of chronic pathology is becoming important for conservation purpose. Chronic gastrointestinal diseases have been shown to have a high incidence in captive population, and it is mainly due to different factors such as diet, living condition and *Helicobacter* spp. infection. In this review, all the studies performed on gastritis in cheetah and factors that can influence chronic pathology are described. These factors are classified in extrinsic (living conditions and diet) and intrinsic (gastric *Helicobacter*-like organisms and immunocompetence). Further research should be aimed to determine risk factors and to individuate strategies to reduce the development of chronic gastrointestinal diseases in captive cheetahs’ population.

## Figures and Tables

**Figure 1 biology-11-00606-f001:**
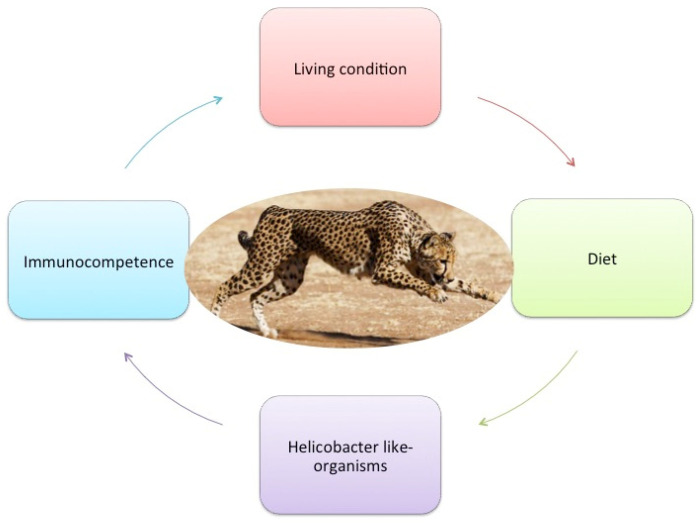
Factors implicated in the pathogenesis of chronic gastrointestinal disorder in cheetah.

**Table 1 biology-11-00606-t001:** Studies about gastroenteric pathologies in cheetahs. No distinction of geographical origin is available, and all animals involved were reported as *Acinonyx jubatus*.

Reference	Aims	No. Cheetah
Eaton et al., 1991 [14]	Gastritis Associated with Gastric Spiral Bacilli in Cheetahs	25
Eaton et al., 1993 [4]	Isolation of *Helicobacter acinonyx* from cheetah with gastritis	4
Munson, 1993 [2]	Diseases of Captive Cheetahs (*Acinonyx jubatus jubatus*) in South Africa	69
Lobetti et al., 1999 [8]	Prevalence of *Helicobacteriosis* and gastritis in Cheetahs	28
Dailidiene et al., 2004 [15]	*Helicobacter pylori*-like gastric pathogen of cheetahs and other big cats	6
Terio et al., 2005 [5]	*Helicobacter* species in captive cheetahs with gastritis	33
Lane et al., 2012 [16]	Effect of diet on the incidence of and mortality owing to gastritis and renal disease in captive cheetahs	72
Terio et al., 2012 [6]	Characterization of the gastric immune response in cheetahs with *Helicobacter*	21
Rossi et al., 2014 [7]	Severe gastritis with double *Helicobacter* spp. infection associated with Barrett’s esophagus in a cheetah	1
Whitehouse-Tedd et al., 2015 [9]	Dietary factors associated with fecal consistency and other indicators of gastrointestinal health in the cheetah	184

## Data Availability

Not applicable.

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
