# Peer review of "Chronic Stress-Related Gastroenteric Pathology in Cheetah: Relation between Intrinsic and Extrinsic Factors"

_biology, 2022, doi:10.3390/biology11040606_

Round 1

Reviewer 1 Report

My comments remain the same from last time, as the author didn't address them. There are only 10 papers included, this seems like more of a summary than a review. I understand more don't exist, but that is therefore not justification for a review.

Author Response

Question 1: My comments remain the same from last time, as the author didn't address them. There are only 10 papers included, this seems like more of a summary than a review. I understand more don't exist, but that is therefore not justification for a review.

Response 1: Thank to the reviewer for the opinion. Authors modified the manuscript making it more suitable for the "review" type. The authors believe that this manuscript may be an addition to the existing bibliography on gastric diseases in cheetah as there isn't a review like this in the bibliography.

Reviewer 2 Report

Chronic stress-related gastroenteric pathology in cheetah: relation between intrinsic and extrinsic factors.

Biology, 1618854

  1. In the paper, it should be clearly emphasized that information on the potential impact of Helicobacters on the physiology and structure of the stomach comes from research conducted on various animal species (usually other than cheetah). Phenomena that, observed e.g. in mice or humans, do not have to take place in the cheetah's stomach. Moreover, so far, different species of Helicobacter have been identified (Terio et al. 2005), and it seems that various species of microorganisms can have different effects on the mucous membrane of the stomach.
  1. Unfortunately, based on our own experience (research was conducted on dogs), it may not be possible to investigate the actual effect of Helicobacters on cheetahs. The main problem may be to find sufficient number of cheetahs in which there will be no Helicobacters in the stomach(“control group”).
  1. There are hard evidence for something that was named “Helicobacter—associated gastritis” in the presented paper? In the study Lobetti et al (1999) among 26 cheetahs included into examined population Helicobacter were present in 23 animals and obvious inflammatory state only in one case. Possibly, Helicobacters (or some species of Helicobacters) can be normal habitants of stomach of cheetahs or gastric pathology develops only in some individuals (for example when immunologic system creates unproper hypersensitivity-like response directed against “physiologic flora of stomach”).

Author Response

Question 1: In the paper, it should be clearly emphasized that information on the potential impact of Helicobacters on the physiology and structure of the stomach comes from research conducted on various animal species (usually other than cheetah). Phenomena that, observed e.g. in mice or humans, do not have to take place in the cheetah's stomach. Moreover, so far, different species of Helicobacter have been identified (Terio et al. 2005), and it seems that various species of microorganisms can have different effects on the mucous membrane of the stomach.

Response 1: Thank to the Reviewer for the comment. We added a sentence to emphasize that the study of the role of Helicobacter has not been as well understood as in other species (line 290-292).

Question 2: Unfortunately, based on our own experience (research was conducted on dogs), it may not be possible to investigate the actual effect of Helicobacters on cheetahs. The main problem may be to find sufficient number of cheetahs in which there will be no Helicobacters in the stomach(“control group”).

Response 2: The review is right, the true role of Helicobacter infection in cheetahs is difficult to investigate due to the number of the population. However, in our recently published scientific article we analyzed the presence of helicobacter in fecal samples of cheetahs with different gastrointestinal symptoms and evaluated the effect of specific probiotics (Mangiaterra et al. Effect of a Probiotic Mixture in Captive Cheetahs (Acinonyx Jubatus) with Gastrointestinal Symptoms-A Pilot Study. Animals (Basel). 2022). We demonstrated that probiotics administration can modulate the gastrointestinal environment, inducing an improvement of symptoms in diseased subjects with Helicobacter spp. infection. Further studies are needed to confirm the role of Helicobacter spp. in cheetahs expanding the study population using molecular biology and histological examination on biopsies.

Question 3: There are hard evidence for something that was named “Helicobacter—associated gastritis” in the presented paper? In the study Lobetti et al (1999) among 26 cheetahs included into examined population Helicobacter were present in 23 animals and obvious inflammatory state only in one case. Possibly, Helicobacters (or some species of Helicobacters) can be normal habitants of stomach of cheetahs or gastric pathology develops only in some individuals (for example when immunologic system creates unproper hypersensitivity-like response directed against “physiologic flora of stomach”).

Response 3: In this review we reported all studies about the possible correlation between the Helicobacter spp. and gastritis in cheetahs based on histological lesion and phenotype of the inflammatory infiltrate. In accordance with the Reviewer 2's comment, the authors suggest the role of other factors in the Helicobacter associated gastritis.

Reviewer 3 Report

Cheetahs are critically endangered, and their lives in the wild and in captivity differ, and a different diet, in particular, may cause different health problems, particularly gastrointestinal diseases in captive cheetahs. Your review article focuses on chronic stress-related gastroenteric pathology in cheetahs-relationship between intrinsic and extrinsic factors, and information on this topic appears to be interesting. I only added a few comments.

Line 14 - the adult population is estimated to be 7100 individuals - please specify where (globally or in Africa?)

This review includes all studies on gastroenteric cheetah pathologies conducted between 1991 and 2021 (line 32-33), but results are only available up to 2015 (line 85-86). Does this imply that there was no additional relevant information?

Is there any newer information in that field because the majority of references are from the late twentieth century, with only a few from the present?

Author Response

Question 1: Cheetahs are critically endangered, and their lives in the wild and in captivity differ, and a different diet, in particular, may cause different health problems, particularly gastrointestinal diseases in captive cheetahs. Your review article focuses on chronic stress-related gastroenteric pathology in cheetahs-relationship between intrinsic and extrinsic factors, and information on this topic appears to be interesting. I only added a few comments.

Response 1: Thank to the review for the suggestion.

Question 2: Line 14 - the adult population is estimated to be 7100 individuals - please specify where (globally or in Africa?)

Response 2: Modified.

Question 3: This review includes all studies on gastroenteric cheetah pathologies conducted between 1991 and 2021 (line 32-33), but results are only available up to 2015 (line 85-86). Does this imply that there was no additional relevant information?

Response 3: Thank to the Reviewer for the comment. There are no studies on gastrointestinal disease in cheetahs after 2015. In the materials and methods, we have included the year 2021 for contemporaneity with the manuscript.

Question 4: Is there any newer information in that field because the majority of references are from the late twentieth century, with only a few from the present?

Response 4: There is no information on the lack of more recent studies. In these last years, our research group has been developing research on gastric pathologies in cheetahs in Europe and Namibia. The last was published a month ago (Mangiaterra et al. Effect of a Probiotic Mixture in Captive Cheetahs (Acinonyx Jubatus) with Gastrointestinal Symptoms-A Pilot Study. Animals (Basel). 2022).